# Effects of Lower Limb Revascularization on the Microcirculation of the Foot: A Retrospective Cohort Study

**DOI:** 10.3390/diagnostics12061320

**Published:** 2022-05-26

**Authors:** Gennady Geskin, Michael D. Mulock, Nicole L. Tomko, Anna Dasta, Sandeep Gopalakrishnan

**Affiliations:** 1Greater Pittsburgh Vascular Associates, Pittsburgh, PA 15025, USA; ggdoc65@gmail.com (G.G.); mmulock@gmail.com (M.D.M.); ntomko@pittvascular.com (N.L.T.); dastaanna@yahoo.com (A.D.); 2College of Nursing, University of Wisconsin-Milwaukee, Milwaukee, WI 53211, USA

**Keywords:** microcirculation, microvasculature, macrovasculature, revascularization, chronic limb-threatening ischemia, perfusion, oxygenation

## Abstract

Background: Current assessment standards in chronic limb-threatening ischemia (CLTI) focus on macrovascular function while neglecting the microcirculation. Multispectral near-infrared spectroscopy (NIRS) provides hemodynamic characteristics of the microcirculation (i.e., capillaries) and may be a powerful tool for monitoring CLTI and preventing extremity loss. The aims of this study were to (1) investigate the effects of lower limb revascularization on the microcirculation and (2) determine if macrovascular and microvascular assessments correlate. Methods: An observational, retrospective cohort study of 38 endovascular interventions in 30 CLTI subjects was analyzed pre- and post-intervention for arterial Doppler acceleration times (AcT; macrovascular) and NIRS metrics (microvascular). Pre-intervention ankle-brachial index (ABI) was also analyzed. Results: AcT significantly decreased (*p* = 0.009) while oxyhemoglobin (HbO) significantly increased (*p* < 0.04) after endovascular intervention, indicating treatment efficacy. However, macrovascular measurements (ABI, AcT) and NIRS metrics of oxygenation and perfusion did not correlate (*p* > 0.06, r^2^ < 0.15, *n* = 23) indicating that macro- and microvascular assessment are not congruent. Conclusion: These findings suggest that macrovascular and microvascular assessments can determine interventional efficacy in their corresponding vasculature. Their lack of correlation, however, suggests the need for simultaneous assessment as independent use may cause diagnostic information to be missed.

## 1. Introduction

Lower extremity peripheral arterial disease (PAD) is estimated to affect >8.5 million Americans above the age of 40 years [1] with an estimated 202 million people worldwide with PAD as of 2013 [2]. A small percentage of PAD patients (1–3%) progress to chronic limb-threatening ischemia (CLTI), a severe complication of PAD, leading to mortality rates that exceed other occlusive cardiovascular diseases such as coronary artery disease [3]. The American Heart Association (AHA)/American College of Cardiology (ACC) guidelines describe CLTI as a condition characterized by chronic (≥2 weeks) ischemic rest pain with non-healing ulcers or gangrene in one or both legs caused by a clinically proven arterial occlusive disease [4]. The mortality rate in CLTI is 20% within 6 months of diagnosis to more than 50% at five years post-diagnosis [5]. CLTI limb preservation prognosis is very poor as more than 200,000 amputations are performed in the United States each year resulting in a substantial burden on healthcare systems and resources [6].

Treatment of CLTI requires a multidisciplinary approach using medical, surgical, and endovascular interventions with the goal of relieving the patient’s symptoms and enhancing quality of life. Apart from modifying the risk factors, revascularization is a key management procedure for treating CLTI patients along with pharmacological interventions (e.g., antihypertensives, antiplatelets) and wound healing therapies [7]. Previous work has shown improved rates of limb salvage following revascularization in patients with PAD compared to other medical treatment options [8]. No-option critical limb ischemia (NO-CLI) is a clinical challenge as the patients are ineligible for surgical or endovascular revascularization. Further, no pharmacological interventions have proven to be effective in treating NO-CLI patients, and quality of life is poor in this population [9].

Multiple clinical tests are used to identify micro and/or macrocirculatory hypoperfusion due to occlusion in CLTI. These tests include assessing limb transcutaneous oximetry, the ankle-brachial index (ABI), skin perfusion pressure, arterial acceleration time (AcT), and the use of phosphorescence biosensors such as indocyanine green [10]. Many other experimental technologies like indigo-carmine angiography, computed tomographic perfusion, contrast-enhanced ultrasound, magnetic resonance angiography, and hyperspectral imaging have been introduced into the clinical practice as investigational techniques to assess tissue perfusion [10]. Current non-invasive perfusion assessments in clinical practice do not meet all clinical needs as only angiographic endpoints are considered reliable in determining the effectiveness of interventions [11]. Multispectral near-infrared spectroscopy (NIRS) imaging may provide unmet clinical utility as it can assess regional tissue hemodynamics and provide spatial information about microvascular tissue oxygenation and perfusion. This imaging modality may be able to inform and support clinicians in determining treatment algorithms, including the revascularization strategy in addition to monitoring and assessing the effectiveness of interventions. Despite the importance of microcirculatory assessments in CLTI, it is unknown how microcirculatory tissue oxygenation parameters derived from multispectral NIRS imaging relate to the macrocirculatory assessments commonly used to identify hypoperfusion. Historically, the term microcirculation has held different connotations as it has been applied to both small (pedal) artery disease and microvascular disease (MVD) [12]. While the small pedal arteries of the foot are 1–2 mm in diameter, MVD is used to describe a range of conditions that impact myogenically-regulated vessels with diameters of ~100 μm (i.e., arterioles, venules) down to ~10 μm (i.e., capillaries) [13]. These classifications, therefore, differ by an order of magnitude and should be nuanced as such. In the context of the present paper, microcirculation will be used to refer to arterioles, venules, and capillaries.

The primary objective of this study was to investigate the effect of lower limb revascularization in patients with PAD on dorsal and plantar microcirculation using multispectral NIRS. A secondary objective was to determine if there is a correlation between changes in macrocirculatory AcT obtained from arterial Doppler ultrasonography and changes in NIRS-derived microcirculatory metrics from pre- to post-limb intervention. We hypothesized that following revascularization, (i) the NIRS-derived microvascular parameters would increase thereby indicating improved microcirculatory perfusion, and (ii) macrocirculatory AcTs would correlate with microcirculatory measures, suggesting that macrovascular assessment is a viable surrogate measure of microcirculatory perfusion.

## 2. Materials and Methods

### 2.1. Demographics

This retrospective study included patients with peripheral arterial disease and non-healing foot ulcers (*n* = 30) who underwent vascular intervention at a specialized vascular center (Greater Pittsburgh Vascular Associates, Pittsburgh, PA, USA) between October 2019 to February 2020. A total of 38 lower limb interventions were performed during this time span. Due to the onset of COVID-19, follow-up assessment was largely incomplete and was not taken into consideration for this present analysis.

### 2.2. Macrocirculatory Assessment

Arterial Doppler ultrasound was used to evaluate medial and lateral tarsal artery acceleration times as previously described [14,15]. In a subset of patients (*n* = 26), an additional macrovasculature assessment of ankle-brachial index (ABI) was taken prior to their interventions. ABIs were performed and assessed according to the guidelines of the American Heart Association [16].

### 2.3. Microcirculatory Assessment

Multispectral near-infrared spectroscopy (NIRS) was used to noninvasively monitor the microcirculation of dermal tissue. Specifically, a Snapshot_NIR_ (KD203, Kent Imaging Inc., Calgary, AB, Canada) system was used to measure tissue oxygen saturation (StO_2_; %), oxyhemoglobin (HbO; a.u.), deoxyhemoglobin (Hb; a.u.), and total hemoglobin (TotHb; a.u.) before and after vascular intervention (within 1 h). These measurements required holding the NIRS camera parallel to the tissue of interest at a distance of 12 inches and then capturing an image. The resultant oxygenation image covers a field-of-view (FOV) of 15 cm × 20 cm. The areas of interest in this study were both the plantar and dorsal surfaces of the lower extremity. Accordingly, two corresponding images were taken for each time point. The metrics reported by the device come from superficial vascular beds (<3 mm) making the readings indicative of oxygenation and perfusion in the arterioles, venules, and capillaries (i.e., microcirculation). An important caveat to consider with penetration depths is that dorsal and plantar circulations have different epidermal thicknesses. This makes it hard to directly compare plantar and dorsal hemodynamics as there can be a discrepancy in how much blood is being sampled in each corresponding location dependent on epidermal thickness.

Data for this study was retrospectively analyzed using a newer software algorithm that is also cleared by FDA and Health Canada (KD204). This newer software algorithm was used as it provided an automatic adjustment for melanin content to more accurately determine oxygenation and perfusion values. Each image was analyzed using four anatomical locations: medial midfoot, medial forefoot, middle/lateral forefoot, and hallux with respective locations for dorsal images. These locations were chosen to give a comprehensive and accurate representation of the oxygenation of the foot while providing a standardized way of collecting data and minimizing light scattering that occurs with tissue that is not parallel to the camera (e.g., lateral foot). While this may not provide an accurate measure of total foot circulation as it neglects lateral arteries (i.e., lateral tarsal artery in the dorsal circulation, lateral plantar artery in the plantar circulation), these selections were chosen for data robustness at the cost of analyzing the whole foot. After the four points had been selected, the oxygenation metrics were averaged from the four sites to give average oxygenation for the entire plantar or dorsal surface within each image. When not all aspects of the foot were present (e.g., hallux amputation), an average of three of the anatomical locations were used. Figure 1 depicts before and after images using NIRS for assessment of dorsal surface tissue oxygen saturation.

### 2.4. Statistical Analysis

Continuous data are represented as means ± standard deviations. Variables were tested for normality of distributions using the Shapiro-Wilk test. A two-way Student’s t-test for pairwise comparisons was used to analyze data before and after the vascular intervention. Pearson correlations were used to determine relationships between macrovascular variables and microvascular variables both prior to intervention as well as delta changes from the intervention. Statistical analysis was conducted using the Python programming language (Python 3.9.5. Python Software Foundation). Statistical significance was set at *p* < 0.05.

## 3. Results

Demographics and comorbidities are presented in Table 1. Briefly, the average age of the patients (*n* = 30) was 72.2 years (range 56–94) while hypertension (80%), dyslipidemia (50%), and diabetes mellitus (43%) were predominant comorbidities within this cohort. Although many risk factors for PAD are shared between both sexes, traditionally it is considered a male-dominated disorder [17], and the patient population in this study was consistent with prior reports (Male:Female, 7:3). Additionally, patients who smoke are at an increased risk of developing symptomatic PAD when compared to non-smokers [17], and our demographics represented 27% of smokers. Note that data collection conditions (e.g., room temperature and lighting, anatomical position for each metric) were not recorded and are subject to variability in this analysis and should therefore be considered as a limitation.

Arterial segment(s) for each intervention and procedure used are outlined in Table 2. Lower limb revascularizations were performed using ablation (45%), atherectomy (37%), and thrombectomy (3%) with angioplasty. The superficial femoral artery (50%) and anterior tibial artery (53%) were the most common revascularized segments. It should then be expected in most cases that the dorsal circulation of the foot be directly affected while the plantar circulation would be indirectly affected [18]. Unifocal (39%) and bifocal (39%) lesions were present in most patients. In the subset of 26 individuals, pre-intervention ABIs were 0.56 ± 0.22.

Figure 2 illustrates the change in macrocirculatory acceleration time from before vascular intervention (193.93 ± 118.41 ms) to after vascular intervention (160.38 ± 92.19 ms; *p* = 0.009). As acceleration times indicate resistive arterial flow or vessel patency, the decreased acceleration time is expected with reduced or removed occlusions.

Figure 3 illustrates the changes in microcirculatory NIRS metrics from before to after vascular intervention. In the plantar circulation, StO_2_ increased from pre-intervention (69.70 ± 9.46%) to post-intervention (71.72 ± 9.98%; *p* = 0.07; Figure 3A). HbO significantly increased from before (0.53 ± 0.14 a.u.) to after (0.57 ± 0.13 a.u.; *p* = 0.04; Figure 3C) vascular intervention. Hb remained constant from pre-intervention (0.22 ± 0.05 a.u.) to post-intervention (0.21 ± 0.06 a.u.; *p* = 0.28; Figure 3E). TotHb significantly increased pre-intervention (0.75 ± 0.11 a.u.) to post intervention (0.78 ± 0.09 a.u.; *p* = 0.04; Figure 3G). In the dorsal circulation, StO_2_ significantly increased from pre-intervention (60.19 ± 12.93%) to post-intervention (64.63 ± 14.36%; *p* = 0.005; Figure 3B). HbO significantly increased from before (0.38 ± 0.13 a.u.) to after (0.42 ± 0.14 a.u.; *p* = 0.007; Figure 3D) vascular intervention. Hb significantly decreased from pre-intervention (0.24 ± 0.06 a.u.) to post-intervention (0.21 ± 0.06 a.u.; *p* = 0.003; Figure 3F). TotHb remained constant from pre-intervention (0.62 ± 0.11 a.u.) to post intervention (0.64 ± 0.10 a.u.; *p* = 0.14; Figure 3H). These findings are consistent with most interventions directly increasing blood flow to the dorsal circulation as the anterior tibial artery was the most common revascularized vessel.

Figure 4 correlates the pre-intervention macrocirculatory metrics of ABI and AcT. There was a moderate negative correlation (r^2^ = 0.41, *p* = 0.002, *n* = 21) between ABI and AcT suggesting that 41% of the variability in AcT can be accounted for with ABI. These findings are in-line with previous work that has shown arterial acceleration time measured in the lateral plantar artery and dorsalis pedis artery to significantly correlate with ABIs and TBIs [14,15].

Figure 5 correlates the pre-intervention macrocirculatory metric of ABI with NIRS microcirculatory metrics. There was no significant correlation between ABI and StO_2_, HbO, or TotHb within either the dorsal or plantar circulations (r^2^ ≤ 0.15, *p* > 0.06; Figure 5A,C,D). As StO_2_ and HbO are indicative of microvascular oxygenation and TotHb is indicative of relative microvascular perfusion, the effects of macrovascular vessel patency were expected to correlate with microvascular perfusion and/or oxygenation. There was no significant correlation between ABIs and Hb in the dorsal circulation (r^2^ = 0.04, *p* = 0.34). There was a weak positive correlation between ABIs and Hb in the plantar surface (r^2^ = 0.18, *p* = 0.04), however, as Hb is a metric of oxygen extraction [19] and not oxygenation or perfusion, these findings merit consideration (see Section 4 Discussion).

Figure 6 correlates the change in macrocirculatory AcTs with changes in NIRS microcirculatory metrics. There was no significant correlation between ∆AcT and any of the NIRS metrics (∆StO_2_, ∆Hb, ∆TotHb, or ∆HbO) in the dorsal circulation (r^2^ < 0.08, *p* > 0.13, *n* = 29; Figure 6). Dorsal NIRS metrics were chosen for correlation as AcTs were taken from the dorsal circulation thereby allowing a direct comparison. For completeness, plantar NIRS metrics and AcTs were compared and were not significantly correlated (*p* > 0.45).

## 4. Discussion

The primary objective of this study was to assess the effects of lower limb revascularization on multispectral NIRS-derived microcirculatory measurements. The results showed significant increases in StO_2_ and HbO indicating increased microvascular oxygenation and increases in total hemoglobin indicating increased microvascular perfusion. These findings highlight the potential utility of multispectral NIRS imaging in determining the effectiveness of revascularization interventions on the microcirculation in PAD. The secondary objective of this study was to determine if relationships exist between macrovascular and microvascular assessments. Despite a good agreement between macrovascular tests (ABI and AcT), no significant relationship was found between NIRS-derived microcirculatory parameters and macrovascular assessment tests. This finding suggests the need to assess macrocirculatory and microcirculatory function independently as using these tests interchangeably may cause diagnostic information to be missed.

CLTI is a major complication of PAD requiring personalized evaluation and treatment protocols as timely revascularization is imperative [10]. It is thus universally accepted that a thorough evaluation of perfusion is important in the diagnosis and management of CLTI [10]. This is often achieved by using arterial duplex/Doppler ultrasound, ABI, toe-brachial index, skin perfusion pressure, or transcutaneous oximetry to monitor arterial perfusion [10]. As the goal of revascularization in CLTI is to reestablish perfusion to ischemic tissue, it is crucial to have accurate assessment tools to continuously monitor hemodynamic changes. However, there appears to be a disarticulation between macrovascular and microvascular assessment as microvascular assessment is not a standard procedure despite the significant health risks associated with MVD [13]. It is therefore concerning that ABI, the universal assessment standard has been shown to not correlate with StO_2_ or other microvasculature assessments in the current study and existing literature [20,21,22]. Interestingly, Chin et al. reported a weak negative correlation between plantar arch deoxyhemoglobin and ABI but they note the lack of clinical significance (9% effect size) [23]. Further, Chin et al. included ABI values > 1.2 with 10 values > 1.5 thereby including non-compressible vessels that drove the negative correlation. As ABI is non-linear above 1.2, this violates correlation assumptions and undermines the validity of these findings. Recent work by Grambow et al. further complicates this as angiosomal assessment found that ABI and StO_2_ only correlated in the lateral plantar and peroneal arteries while ABI and a NIRS perfusion index only correlated in the medial plantar artery and sural arteries in 17 patients who underwent superficial femoral artery endovascular treatment [24]. It should also be further noted that effect sizes were small (r^2^ < 0.29) with borderline statistical significance in all cases (*p* > 0.025) which may give rise to type 1 errors. Their lack of correlation and non-significant changes in StO_2_ in some studied angiosomes [24] suggest that changes in perfusion may take weeks after respective therapy to become apparent [25] and further validates the need to assess both macrovasculature and microvasculature. These findings combined with the findings of macrovascular agreement in the present study (Figure 4) and previous reports [14,15] support the argument that macro- and microvasculature status may not be synonymous. These differences from macro- to micro- might be explained by ischemic reperfusion injury, initial arteriosclerotic occlusion of the respective terminal arteries, or a temporal delay in function [24].

Macrovascular (e.g., coronary artery disease, PAD) and microvascular (e.g., diabetic nephropathy, neuropathy, retinopathy, microangiopathy) dysfunction are both common in patients with diabetes where prolonged exposure to hyperglycemia causes chronic cellular stress (e.g., osmotic, oxidative) [26]. This chronic cellular stress elicits an increase in thickness of the capillaries within the dermal circulation along with a decrease in capillary lumen diameter and reduced perfusion capacity [27,28]. Damage to the endothelium results in an altered vascular tone which promotes atherosclerosis [29]. Thickening of the arterioles and capillaries is akin to atherosclerosis of the larger vessels thereby impairing healthy vasomotor function in the cutaneous microvasculature. This in turn leads to MVD which is associated with 18% of all amputations and 15% of all below-the-knee amputations [13]. While most common in diabetes, MVD is not exclusively found in diabetics, and targeting diabetes in isolation may miss opportunities for preventing amputation in a larger population at risk [13]. Further, although amputation risk among those with MVD is independent of those with PAD, MVD likely participates in the development of adverse limb events in PAD [13]. This is troubling as MVD is limited or forgotten in the current framework of critical limb ischemia [13,30]. This opens the door for the assessment of both large conduit vessels and microcirculation to determine whether interventional revascularization will improve outcomes.

The present study is in line with previous research [20,21,22] as ABI did not correlate with perfusion and oxygenation metrics of StO_2_, HbO, or TotHb. While ABI did correlate with Hb (Figure 5B), it should be interpreted with caution as Hb is not a marker of perfusion but of oxygen extraction [19]. We suggest there are multiple possible explanations for this relationship: (1) impaired oxygen extraction in disease, (2) heterogeneity of oxygen delivery and MVD, and (3) anatomic arteriovenous shunting of blood. Previous work in disease has shown that increased delivery does not boost oxygen consumption [31] and critical illness appears to be associated with impaired oxygen extraction [32,33]. Note that the relationship between oxygen delivery and oxygen consumption is biphasic [34] such that consumption is dependent on delivery to a point. As we found no correlation between total hemoglobin (surrogate of relative perfusion) and ABI, there would seemingly be other factors involved than just depressed oxygen consumption as a product of depressed oxygen delivery. Given our varied comorbidities (see Table 1), it may be reasonable to assume impaired oxygen extractions in some cases, especially with increased disease severity. Next, heterogeneity of oxygen delivery can be described as a mismatch of oxygen demand to oxygen supply [34]. Combined with MVD, it is possible that decreased capillary luminal diameters [27,28] lead to heterogeneity of blood flow [34] thereby impairing oxygen extraction. It has also been suggested that oxygen metabolism is impaired by MVD [35] which would similarly decrease oxygen extraction and lower deoxyhemoglobin amounts. Another possible explanation is that anatomical arteriovenous anastomoses shunting contributes by allowing oxygen to bypass the capillary networks in diabetics [36,37]. As the ABI and deoxyhemoglobin relationship is small (18% effect size) and the present data is limited, a definitive mechanism cannot be elucidated and warrants future investigation.

### 4.1. Vessel Intervention Variability

The anatomy of the foot offers redundancy to blood flow via anastomoses and perforating vessels. This allows perfusion of tissue to be directly or indirectly affected depending on upstream vessel patency. Kawarda et al. [18] found that single tibial arterial revascularization increased microcirculation by similar amounts on both dorsal and plantar surfaces which agrees with prior work showing no clinical difference between direct and indirect revascularization [38,39]. As this study included various arterial segments, this highlights two important considerations. First, infrapopliteal revascularizations increase oxygenation and perfusion to both dorsal and plantar surfaces either directly or indirectly. However, this is contingent on the degree of vascular occlusion throughout the circulation of the foot and whether direct and indirect accesses are viable. This may in part explain the differences between the hemodynamics of the dorsal and plantar surfaces (Figure 3) as the current cohort preferentially had more anterior interventions that may point to residual occlusion within the dorsal circulations and, therefore, reduced perfusion and oxygenation readings. Second, this study involved interventions above the popliteal artery. As those above the knee revascularizations increased patency, the below the knee vessels would need to also be unobstructed to increase foot perfusion and oxygenation. Given the limited sample in this study with varying degrees of post-surgical vessel patency, the degree of oxygenation and perfusion increases from revascularization cannot be extrapolated to a larger population at this time.

### 4.2. Simultaneous Assessment of Oxygenation and Perfusion

The present study ties into previous work demonstrating the importance of microvascular assessment in determining CLI revascularization efficacy [40]. Yu et al. used a similar NIRS device to monitor the acute effects of macro-revascularization on microvascular oxygenation and perfusion intra- and perioperatively [40]. They found that revascularization significantly and acutely increased blood flow post-intervention, but there was not a similar increase in oxygenation as expected. This observation likely arises from increased oxygen delivery being counterbalanced by oxygen utilization in the ischemic tissue immediately after the intervention. This discrepancy between oxygenation and perfusion postoperatively [24,40] highlights the importance of simultaneous monitoring of both parameters on an ongoing basis. While hemodynamic characteristics during and following revascularization have shown prognostic value, future work looking at long-term outcomes and hemodynamic changes is warranted.

Many perfusion assessment tools are used to assess CLI in clinical practice, but no modality has been able to predict wound healing trajectory or amputation incidence [41]. This is further complicated as pathophysiological challenges (e.g., calcinosis, non-compressible vessels) can limit the utility of vascular tests like ABI and toe-brachial index [10]. While guidelines and expert opinions provide a framework for the assessment of perfusion, more precise and accurate intraprocedural tools are needed to quantitatively assess perfusion and oxygenation as current tools are limited (e.g., angiography) [10]. Fast, accurate, and non-invasive techniques that can quantify tissue perfusion and/or oxygenation would be an invaluable asset in evaluating revascularization efficacy. This study demonstrates the potential of NIRS to be used for perioperative hemodynamic monitoring while also illuminating the importance of microcirculatory evaluation.

## 5. Conclusions

A retrospective analysis of 30 chronic limb-threatening ischemia patients was conducted to assess lower limb perfusion and oxygenation pre- and post-endovascular intervention. Findings suggest macrocirculatory and microcirculatory assessment should not be used interchangeably as this may cause diagnostic information to be missed. Synergistic assessment of both macrocirculation and microcirculation is warranted as they provide complementary information on perioperative hemodynamic monitoring.

## Figures and Tables

**Figure 1 diagnostics-12-01320-f001:**
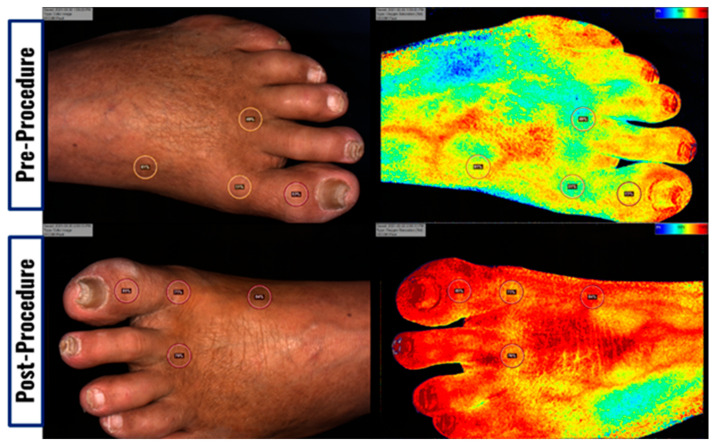
Clinical images (**left**) and tissue oxygen saturation (StO_2_) images (**right**) of pre-vascular intervention (**top**) and post-vascular intervention (**bottom**) in an 80-year-old diabetic male who had atherectomy and ablation of anterior and posterior tibial arteries. StO_2_ of the foot is represented as a spectrum of 100% (red) to 0% (blue). Images were taken using Snapshot_NIR_ (KD203, Kent Imaging Inc., Calgary, AB, Canada).

**Figure 2 diagnostics-12-01320-f002:**
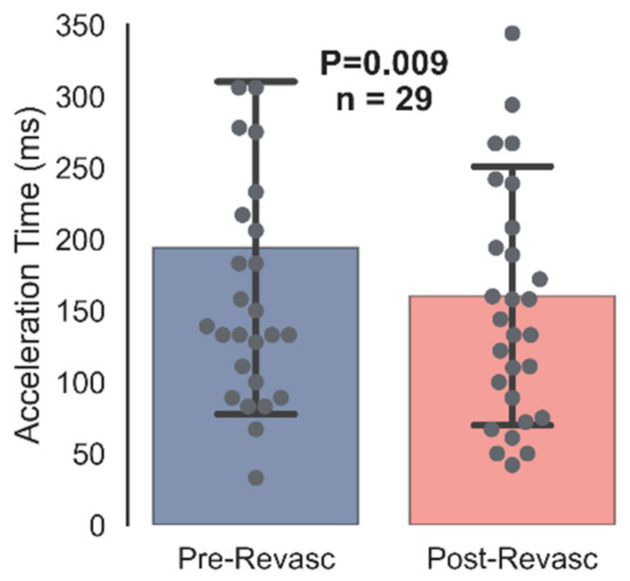
Before and after vascular intervention comparison of AcT. AcT measures were taken from the medial side of the revascularized foot via arterial duplex ultrasound. AcT, acceleration time. Presented as means ± SD. *n* = 29.

**Figure 3 diagnostics-12-01320-f003:**
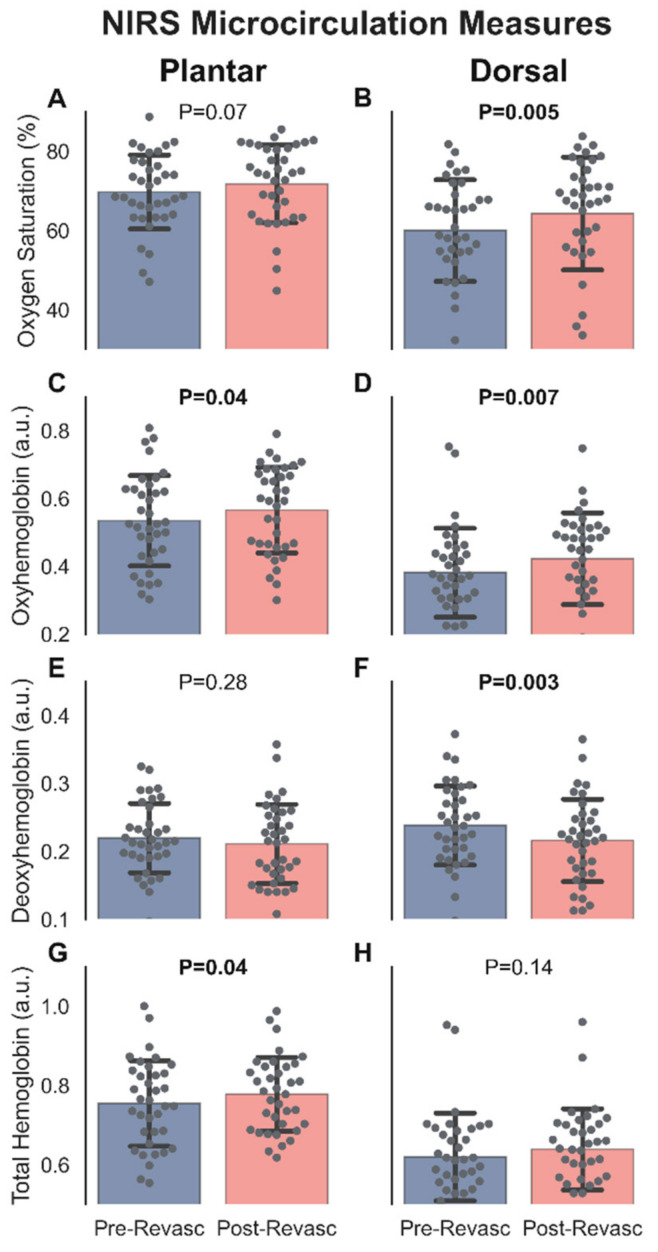
Before and after vascular intervention comparisons of tissue oxygen saturation (**A**,**B**), oxyhemoglobin (**C**,**D**), deoxyhemoglobin (**E**,**F**), and total hemoglobin (**G**,**H**) in the plantar (**left**) and dorsal (**right**) microcirculations. Microcirculatory measures were taken using Snapshot_NIR_, a near-infrared spectroscopy camera. Presented as means ± SD. Plantar: *n* = 36. Dorsal: *n* = 37.

**Figure 4 diagnostics-12-01320-f004:**
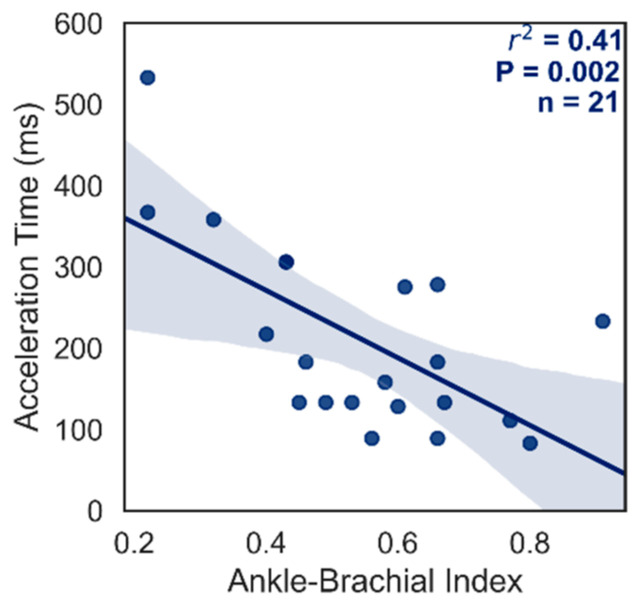
Relationship between AcT and ABI macrocirculatory metrics prior to vascular intervention. Pearson correlation values for r^2^, *p*, and *n* are indicated on the graph. Results are comparable to previously published work [14,15]. AcT, acceleration time; ABI, ankle-brachial index.

**Figure 5 diagnostics-12-01320-f005:**
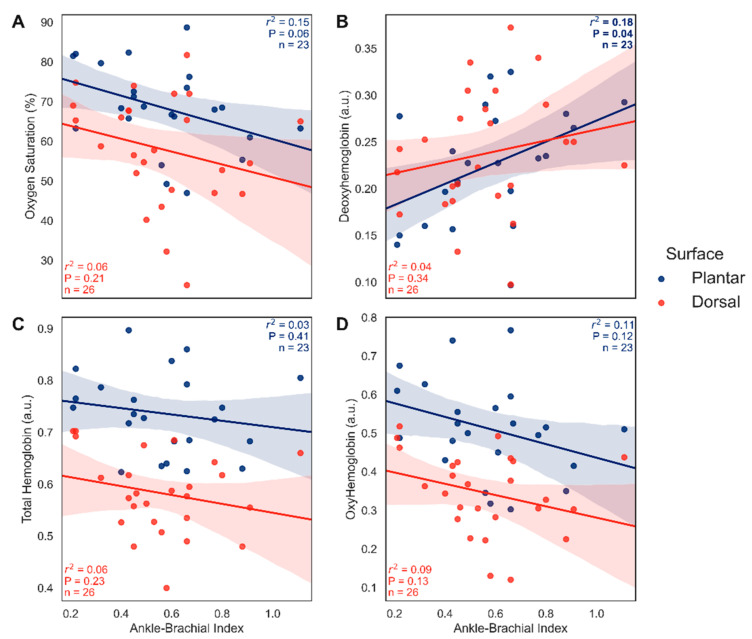
Relationship between ABI and NIRS microcirculatory metrics prior to vascular intervention. (**A**) relationship between ABI and tissue oxygen saturation (StO_2_; %). (**B**) relationship between ABI and deoxyhemoglobin (Hb; a.u.). (**C**) relationship between ABI and total hemoglobin (TotHb; a.u.). (**D**) relationship between ABI and oxyhemoglobin (HbO; a.u.). Respective r^2^, *p*, and *n* are indicated on each panel with 95% confidence intervals. ABI, ankle-brachial index; NIRS, near-infrared spectroscopy.

**Figure 6 diagnostics-12-01320-f006:**
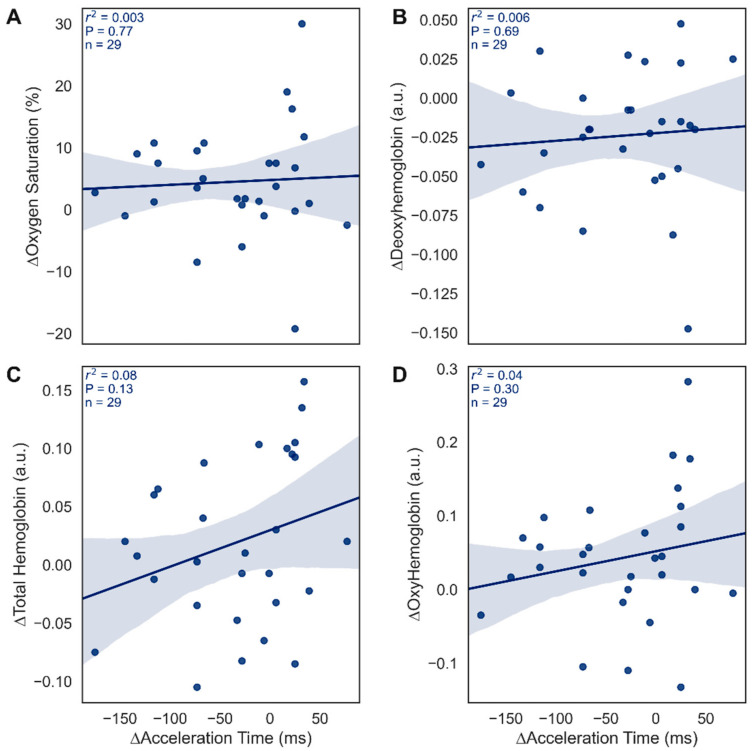
Relationship between changes in AcT and NIRS microcirculatory metrics from pre- to post-vascular intervention. (**A**) relationship between AcT and tissue oxygen saturation (StO_2_; %). (**B**) relationship between AcT and deoxyhemoglobin (Hb; a.u.). (**C**) relationship between AcT and total hemoglobin (TotHb; a.u.). (**D**) relationship between AcT and oxyhemoglobin (HbO; a.u.). Respective r^2^, *p*, and *n* are indicated on each panel with 95% confidence intervals. AcT, acceleration time; NIRS, near-infrared spectroscopy.

**Table 1 diagnostics-12-01320-t001:** Subject demographics and comorbidities.

Demographics	Patients (*n* = 30)	
Age (years)	72.2	(56–94)
Sex (male/female)	21/9	
**Comorbidities**	**# and % of Cases**	
Smoking	8	27%
Hypertension	24	80%
Dyslipidemia	15	50%
Diabetes mellitus	13	43%
Coronary artery disease	8	27%
Congestive heart failure	2	7%
Liver transplant	1	3%
Chronic obstructive pulmonary disease	1	3%

**Table 2 diagnostics-12-01320-t002:** Endovascular intervention summary.

Intervention with Angioplasty (*n* = 38)	# and % of Cases	
Ablation	17	45%
Atherectomy	14	37%
Thromectomy	1	3%
**Arterial Segment**	**# and % of Cases**	
Common iliac artery	4	11%
External iliac artery	6	16%
Common femoral artery	4	11%
Superficial femoral artery	19	50%
Popliteal artery	4	11%
Tibioperoneal trunk	3	8%
Peroneal artery	2	5%
Posterior tibial artery	6	16%
Anterior tibial artery	20	53%
Dorsalis pedis	4	11%
Lateral plantar artery	1	3%
**Lesions in Each Intervention**	**# and % of Cases**	
Unifocal	15	39%
Bifocal	15	39%
Trifocal	4	11%
Quadfocal	3	8%
Quintfocal	1	3%

## Data Availability

The deidentified data that support the findings of this study are available from the corresponding author upon reasonable request from a qualified researcher.

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
