# Peer review of "Effects of Lower Limb Revascularization on the Microcirculation of the Foot: A Retrospective Cohort Study"

_diagnostics, 2022, doi:10.3390/diagnostics12061320_

Round 1
Reviewer 1 Report
The paper requires a minimal comparison with acute situations, even in the ischemia generated from the SARS-COV-2 infection:
Barac S, Onofrei RR, Neagoe PV, Popescu AI, Pantea S, Rață AL. An
Observational Study on Patients with Acute Limb Ischemia and SARS-CoV-2
Infection: Early and Late Results in Limb Salvage Rate. J Clin Med. 2021 Oct
29;10(21):5083. doi: 10.3390/jcm10215083. PMID: 34768611; PMCID: PMC8584433.
You must also take into consideration an experimental model for the evaluation of tissue perfusion before and after a revascularization procedure, on an animal model. See citation: Hoinoiu B, Jiga LP, Nistor A, Dornean V, Barac S, Miclaus G, Ionac M, Hoinoiu T. Chronic Hindlimb Ischemia Assessment; Quantitative Evaluation Using Laser
Doppler in a Rodent Model of Surgically Induced Peripheral Arterial Occlusion. Diagnostics (Basel). 2019 Oct 2;9(4):139. doi: 10.3390/diagnostics9040139. PMID:31581692; PMCID: PMC6963965.
You must have a clear delimitation of the Conclusion section.
Author Response
Response 1: Thank you for your response. For the sake of this study, we feel that adding these two studies are outside the scope of this paper and distract from our overall purpose.
In Barac et al., 2021, the study is focused on amputation outcomes and thrombotic events. As acute limb ischemia arises from hypercoagulability in COVID-19, it differs from chronic limb threatening ischemia which occurs because of proliferation of foam cells into the endothelial linings of vessels. Further, there is no analysis of perfusion in Barac et al., 2021, but instead of blood biomarkers which is periphery to our work in perfusion and does not provide focused insight. While both studies share the commonality of open surgery or endovascular intervention for revascularization, that is where the commonalities end.
In suggesting Hoinoiu et al., 2019 for an experimental model of perfusion before and after intervention, this again seems off-topic as there is extensive work in human research to assess perfusion (see references 18, 22, 24; PMID: 25813533, 25308948) which makes a more meaningful comparison than work relating to animal models. Note also that we cite the American Heart Association (reference 10) with their consensus guidelines for perfusion assessment which is a very recent and comprehensive synthesis of the literature while also providing human work on contrast imaging and CT perfusion. Taken together, Hoinoiu et al., 2019 does not provide enhanced understanding of our current work.
While Barac et al., 2021 and Hoinoiu et al., 2019 both contribute to their respective fields, it should also be noted that there are numerous issues with these studies such as incorrect statistical analysis in Hoinoiu et al., 2019 (i.e., using Mann-Whitney U tests instead of Friedman’s test as a non-parametric repeated measures ANOVA, not using post-hoc corrections (e.g., Bonferroni) for multiple comparisons) and drawing conclusions based off of underpowered data in Barac et al., 2021 (e.g., talks about correlation between ferritin levels and mortality while no correlational analysis was undertaken) that deter us from citing these papers.
We have added a clear delimitation of the conclusion as requested.

Reviewer 2 Report
In this study Geskin and collegues aimed at describing the microvascular parameters throughout recovery in the foot of patients with chronic limb-threatening ischemia throughout revascularization surgeries. Secondarily, the authors also aimed at correlating microvascular perfusion parameters with well-established parameters of macrovascular structure and function.
A sample of 30 patients was studied before and after revascularization. Multispectral near-infrared spectroscopy (NIRS) was used to assess microcirculation, whereas arterial Doppler acceleration times and ankle-brachial index assessed macrocirculation. Authors reported significant improvement of microvascular parameters after revascularization, indicating clinical efficacy. However, authors also reported an overall lack of statistical correlation between micro- and macrovascular parameters, suggesting that these vascular compartments should be assessed together to provide a more complete clinical picture.
Overall the study was well-designed and carried out, and the manuscript is well-written. The reported results, despite not being very original, contribute to support the corollary that microvascular assessment is somewhat cumbersome, but should be carried out nonetheless in association with the study of macrocirculation.
Despite their merits, authors should better describe the conditions on which they carried out the macro- and microvascular measurements, namely the room conditions (temperature, lighting), the anatomical position of the patient, the time elapsed since revascularization surgery, etc.
Also, the authors should provide some explanations as to why macro- and microvascular parameters did not correlate well or at all, which would improve the text. They should include in their discussion an explanation for the substantial differences in results between dorsal and plantar circulation. Since NIRS assesses dermal microcirculation the authors should focus on differences in sympathetic innervation and in density of arteriovenous anastomoses. On a side note, perhaps the relation between macro- and microcirculation can be better assessed in the future with dynamic tests (e.g. leg elevation) as well as with turning NIRS into a continuous acquisition technique (e.g. instead of acquiring a single image, acquire a movie clip consisting of several images over time).
A few minor issues should also be corrected:
- “doppler” is a surname, so it should be capitalized (Doppler);
- Lines 71-73 – although of low importance for this particular study, the definition of microcirculation present in this sentence is merely structural. Authors should keep in mind that microcirculation also encompasses vessels with myogenic activity. Therefore, I suggest the authors revise this sentence as to the definition of microcirculation;
- Line 81 – perhaps “hypothesized” (past tense) would be more appropriate;
- Line 95 – perhaps “patients” or “subjects” would be more appropriate terms;
- Lines 117-118 - While the authors are aware of the technical limitations of NIRS, it does not seem legitimate to assume that the chosen regions provide an accurate representation of the oxygenation of the whole foot - the studied regions are supplied by the dorsalis pedis artery, whereas the lateral tarsal artery is also an important artery in supplying the lateral foot;
- Figure 1 - Please describe how tissue oxygenation and image color are related. Also, please indicate sex and age of subject;
- Lines 137-138 – the website is not necessary;
- Table 1 – perhaps "Smoking" is more appropriate than “Tobacco use” since the latter could refer, in a broad context, to several types of tobacco use;
- Line 201 – in “r2<0.15” the symbol should be “equal or less”, considering that in 5A r2 is equal to 0.15;
- Line 220-221 - to state that these variables were correlated when p value is below significance is somewhat misleading. It is preferable to state that the variables were not significantly correlated.
Author Response
Reviewer: 2
In this study Geskin and collegues aimed at describing the microvascular parameters throughout recovery in the foot of patients with chronic limb-threatening ischemia throughout revascularization surgeries. Secondarily, the authors also aimed at correlating microvascular perfusion parameters with well-established parameters of macrovascular structure and function.
A sample of 30 patients was studied before and after revascularization. Multispectral near-infrared spectroscopy (NIRS) was used to assess microcirculation, whereas arterial Doppler acceleration times and ankle-brachial index assessed macrocirculation. Authors reported significant improvement of microvascular parameters after revascularization, indicating clinical efficacy. However, authors also reported an overall lack of statistical correlation between micro- and macrovascular parameters, suggesting that these vascular compartments should be assessed together to provide a more complete clinical picture.
Overall the study was well-designed and carried out, and the manuscript is well-written. The reported results, despite not being very original, contribute to support the corollary that microvascular assessment is somewhat cumbersome, but should be carried out nonetheless in association with the study of macrocirculation.
Response 2: We thank you for your excellent synopsis of our work and your kind words.
Despite their merits, authors should better describe the conditions on which they carried out the macro- and microvascular measurements, namely the room conditions (temperature, lighting), the anatomical position of the patient, the time elapsed since revascularization surgery, etc.
Response 3: Thank you for your input as indeed, this is important information to note. Owing to this being a retrospective analysis of data that was collected over two years ago, addressing some of these considerations is difficult. To address this, we have included a note at the beginning of the results identifying room temperature and lighting then anatomical position as being limitations for this study. For time elapsed since revascularization surgery, a note of the images being taken within 1 hour of the procedure has been added to the microcirculatory assessment section. We hope this is sufficient for the retrospective analysis of the data.
Also, the authors should provide some explanations as to why macro- and microvascular parameters did not correlate well or at all, which would improve the text. They should include in their discussion an explanation for the substantial differences in results between dorsal and plantar circulation. Since NIRS assesses dermal microcirculation the authors should focus on differences in sympathetic innervation and in density of arteriovenous anastomoses. On a side note, perhaps the relation between macro- and microcirculation can be better assessed in the future with dynamic tests (e.g. leg elevation) as well as with turning NIRS into a continuous acquisition technique (e.g. instead of acquiring a single image, acquire a movie clip consisting of several images over time).
Thank you for your commentary and we agree. Without drawing out an already long discussion, we have included the following from lines 283-285. “These differences from macro- to micro- might be explained by ischemic reperfusion in-jury, initial arteriosclerotic occlusion of the respective terminal arteries, or a temporal delay in function.” We hope this is suitable.
For the substantial differences in results between dorsal and plantar circulations, we have added two aspects: 1) the concept of residual occlusion within the dorsal circulation (as a product of the majority of interventions being anterior tibial) that would reduce our measured perfusion and oxygenation for the dorsal circulation (Figure 3) and; 2) commentary on epidermal thicknesses and how much blood is being samples in the dorsal and plantar circulations via NIRS. Together these two points are known limitations with the study that might cause the discrepancy without getting into sympathetic innervation and density of arteriovenous anastomoses which would be hard to quantify or reconcile with a spectrum of diabetic patients who would conceivably have a broad range of innervation and active arteriovenous anastomoses. We have also added the word “anastomoses” to line 321 to highlight the role that arteriovenous anastomotic shunting places in diabetes.
Thank you for the side note as it is something we have been considering. For the leg elevation, a study analysis of pre and post leg raise both before and after revascularization would be a wonderful way to tease apart microcirculatory function as it would introduce another functionality test. In Grambow et al., 2022 (reference 24), they note that changes in perfusion make take weeks to become apparent while they themselves found very little difference at the 1 day and 3 day post surgery timepoint (while they collected day 5, it is not presented). Together, perhaps a leg elevation task would be able to elicit a quantifiable response that would improve out understanding.
As for a continuous acquisition technique, challenges arise in computational power, resolution, portability, and sensitivity of camera-to-tissue distance that may prohibit such forms of NIRS device at the current time. While this NIRS device provides a large swath of area by it a non-contact camera, other surface NIRS devices do provide continuous recording and can be used for longitudinal analysis which is common in plastic and reconstructive surgery (PMID: 29308105, 17532278, 22131106). Feasibility being how it is currently; we welcome a non-contact continuous NIRS device but we might have to wait for that technology to develop.
A few minor issues should also be corrected:
- “doppler” is a surname, so it should be capitalized (Doppler);
Thank you for catching this. Fixed it in the 4 instances in the paper.
- Lines 71-73 – although of low importance for this particular study, the definition of microcirculation present in this sentence is merely structural. Authors should keep in mind that microcirculation also encompasses vessels with myogenic activity. Therefore, I suggest the authors revise this sentence as to the definition of microcirculation;
We are confused about the nuance that would correctly address this omission. Our understanding is that myogenic activity is present in arterioles and venules which are encompassed in this definition. We have therefore added, “myogenically-regulated” to describe the arterioles and venules as a way to address this concern.
- Line 81 – perhaps “hypothesized” (past tense) would be more appropriate;
Fixed
- Line 95 – perhaps “patients” or “subjects” would be more appropriate terms;
Changed to “patients”
- Lines 117-118 - While the authors are aware of the technical limitations of NIRS, it does not seem legitimate to assume that the chosen regions provide an accurate representation of the oxygenation of the whole foot - the studied regions are supplied by the dorsalis pedis artery, whereas the lateral tarsal artery is also an important artery in supplying the lateral foot;
Thank you for this helpful commentary. We have included the following as a solution:
“While this may not provide an accurate measure of total foot circulation as it neglects lateral arteries (i.e., lateral tarsal artery in the dorsal circulation, lateral plantar artery in the plantar circulation), these selections were chosen for data robustness at the cost of analyzing the whole foot.”
- Figure 1 - Please describe how tissue oxygenation and image color are related. Also, please indicate sex and age of subject;
Changed figure legend to the following:
“Clinical images (left) and tissue oxygen saturation (StO2) images (right) of pre-vascular intervention (top) and post-vascular intervention (bottom) in a 80-year-old diabetic male patient who had atherectomy and ablation of anterior and posterior tibial arteries. Oxygenation of the foot is represented as a spec-trum of 100% (red) to 0% (blue). Images were taken using SnapshotNIR (KD203, Kent Imaging Inc., Calgary, AB, Canada).”
- Lines 137-138 – the website is not necessary;
Removed
- Table 1 – perhaps "Smoking" is more appropriate than “Tobacco use” since the latter could refer, in a broad context, to several types of tobacco use;
Changed
- Line 201 – in “r2<0.15” the symbol should be “equal or less”, considering that in 5A r2 is equal to 0.15;
Changed. When we ran statistics, we had the information to multiple extra digits that rounded up to 0.15 and the equality sign was carried forth from the rounding. Thank you for catching this.
- Line 220-221 - to state that these variables were correlated when p value is below significance is somewhat misleading. It is preferable to state that the variables were not significantly correlated.
Changed to, “For completeness, plantar NIRS metrics and AcTs were compared and were not signifi-cantly correlated (P > 0.45).”
Round 2
Reviewer 1 Report
ok